# Impact of COVID-19 Vaccination Status and Confidence on Dietary Practices among Chinese Residents

**DOI:** 10.3390/foods11091365

**Published:** 2022-05-08

**Authors:** Zhongyu Li, Yidi Ma, Shanshan Huo, Yalei Ke, Ai Zhao

**Affiliations:** 1Vanke School of Public Health, Tsinghua University, Beijing 100091, China; zhongyu.li@tufts.edu; 2Friedman School of Nutrition Science and Policy, Tufts University, Boston, MA 02111, USA; 3School of Public Health, Peking University, Beijing 100191, China; 1810306135@pku.edu.cn (Y.M.); tuanzishan@pku.edu.cn (S.H.); keyalei@pku.edu.cn (Y.K.)

**Keywords:** COVID-19, pandemic, vaccine, vaccine confidence, nutrition, dietary behavior, dietary diversity

## Abstract

Healthy diets promote immune functions and have been shown to reduce COVID-19 severity. In 2021, COVID-19 vaccines have become available to the general public. However, whether vaccination status could affect individual and populational health behaviors is unknown. This study aimed to investigate the impacts of vaccination status and confidence on dietary practices. An online survey was conducted in August 2021. We collected data on dietary intake, diversity and behaviors, vaccination status and confidence and socio-demographic characteristics. Among the 5107 responses received, a total of 4873 study participants were included in the final analysis. Most of our participants aged between 18 and 45 years and 82% of them were fully vaccinated against COVID-19. Household level dietary diversity was found to be higher among people who were fully vaccinated (β = 0.321, 95%CI: 0.024 to 0.618) or who were more confident in the protectiveness of the vaccine (β for tertile 3 comparing with lowest tertile = 0.544, 95%CI: 0.407, 0.682). Vaccination promoted the intake of seafood, but it was also positively associated with the consumption of sugar, preserved, fried and barbequed foods and reduced vegetable intake. Higher vaccination confidence was associated with increased consumption of seafood, bean, fruits and vegetables and reduced fat intake. Changes in dietary behaviors compared with early 2021 (when vaccination was not common) were observed and differed by vaccination status and confidence level. Conclusion: COVID-19 vaccination status and confidence had varied, and possibly negative, impacts on dietary intake and behaviors. Our results suggest that vaccination status and confidence might be significant influencing factors affecting people’s health behaviors and highlight that healthy eating should be consistently promoted to prevent poor dietary practices during global health crisis.

## 1. Introduction

The 2019 coronavirus disease (COVID-19) is a severe acute respiratory syndrome caused by SARS coronavirus 2 (SARS-CoV-2) [1]. By February 2022, over 420 million cases and 5.8 million deaths have been recorded worldwide [2]. The COVID-19 pandemic has drastically changed the lives of individuals and resulted in long lasting negative social impacts [3,4]. In early 2021, with the development and mass production of the SARS-CoV-2 vaccine—also called the COVID-19 vaccine—at an unprecedented speed, vaccination has become widely available [5,6]. At the time of this study (August to September 2021), about 67% of the Chinese and 54% of American populations have completed the initial vaccination protocols [7]. As more people became vaccinated throughout the year of 2021, COVID-19 cases dropped, followed by eased pandemic restrictions such as the reopening of restaurants and indoor events. However, the surge in new cases of the Delta variant in late September and the Omicron variant in December 2021 has led to fluctuations in some loosened restrictions, thus extending the influence of the current pandemic and disruptions in daily routines and social activities [8,9].

Changes in dietary behaviors such as eating, cooking and shopping patterns have been observed globally and reported by many studies during the COVID-19 pandemic [10,11,12,13]. Previous studies have shown that alternations in these dietary behaviors are rarely unidirectional and are impacted by socio-demographic factors such as age, gender, education and existing medical conditions such as obesity and type 2 diabetes [14,15,16], indicating distinctive dietary behaviors in response to the pandemic. For instance, a longitudinal study by Dicken et al. revealed that while about 40–55% of study participants decreased consumption of high fat, salt and sugar (HFSS) snacks, 30–55% of people consumed more HFSS snacks in 2020 [17]. Likewise, although people may cook at home more often during the pandemic than they used to, they are not necessarily eating healthier because they may also snack and eat out of boredom or stress more often [10].

Despite the demonstrated effectiveness of COVID-19 vaccines in preventing infection and severe cases, a healthy lifestyle including maintaining a heathy weight, being physically active and having balanced and nutritious diet remain critical in sustaining proper immune system functions [18,19,20]. It has been shown that vitamins, minerals and omega-3 fatty acids can support immune responses and reduce COVID-19 severity and recovery time [12,21,22]. A study among frontline healthcare providers suggested that those who followed a whole food plant-based diet or pescatarian diet—these diets usually contain high amounts of fruits, vegetables and legumes—were less likely to have moderate to severe COVID-19 cases [23]. Moreover, being nutritionally adequate may enhance immune responses to vaccines against infections, therefore highlighting the importance of healthy eating as COVID-19 vaccines have been made available to the general public [24,25].

Very few studies have investigated how COVID-19 vaccination status and confidence affect dietary practices during a pandemic. In this study, we aim to explore the dietary practices after vaccination has become widely available and to evaluate how the vaccination status and confidence could affect people’s dietary practices.

## 2. Materials and Methods

### 2.1. Study Design and Participants

A cross-sectional survey was conducted through the Wenjuanxing e-questionnaire platform (Wenjuan xing Tech Co., Ltd., Chansha, China) in August 2021 to assess dietary diversity and behaviors including food acquisition and intake post COVID-19 vaccine in mainland China. This study is the third wave of a longitudinal nutrition study initiated in March 2020 (the first survey) [26]. Similar to the previous two surveys [26,27], this survey employed a multistage sampling method to identify study participants from different areas of China at first, which ensured the inclusion of people living in South, North and Central China, and then recruited more by a “snowball sampling” method.

To improve data quality, we embedded attention checking questions (e.g., Have you answered the questions seriously? Please choose “rare” in this question) in the online questionnaire to identify participants who likely randomly selected answers. People who failed these attention checking questions were not included in this study. We received a total of 5107 responses by the end of August 2021. Among these, we excluded people who were younger than 18 years old (N = 157), who were not living in mainland China at the time of survey (N = 8) and who completed the survey in less than three minutes (N = 71). Finally, data of 4873 participants were used in subsequent analysis. The geographic distribution of participants is shown in Figure 1.

### 2.2. Data Collection

The e-questionnaire included 30 main questions grouped into five sections: (1) food acquisition, dietary diversity and dietary intake frequency and preferences; (2) changes in specific dietary behaviors; (3) self-reported COVID-19 vaccination status and confidence; (4) chronic disease status; and (5) basic socio-demographic information. There are five possible scenarios for vaccination status: fully vaccinated, received the first dose of the two-dose vaccine, received the first dose of the three-dose vaccine, received the second dose of the three-dose vaccine and not vaccinated. We categorized the above five scenarios into three groups: fully vaccinated, partially vaccinated (completed some but not all required doses) and not vaccinated. Moreover, we asked participants to rate how protective they believed the vaccines were against COVID-19 infection or severe cases on a scale of 0 to 100, and then grouped them into tertiles.

To assess household level dietary diversity, we used the Household Dietary Diversity Score (HDDS, consisting of 12 food groups) to measure household access to food [29], including cereals, roots and tubers, vegetables, fruits, meat, poultry and offal, eggs, fish and seafood, pulses, legumes and nuts, dairy products, oils and fats and sugar and honey and the miscellaneous group (such as condiments, snacks and beverages). The same food groups adopted from the HDDS survey except the miscellaneous group were also used to evaluate dietary intake frequencies in the past month. Participants were allowed to choose from one of the five options: (1) less than once per week; (2) 1–3 times per week; (3) 4–6 times per week; (4) once per day; and (5) more than once per day for each of the 12 food items. We additionally assessed the frequencies of consuming barbequed food, preserved food and fried food, using the same approach described above.

To explore changes in specific dietary behaviors such as the frequencies of eating out, cooking at home, grocery shopping online and consumption of frozen food, we asked participants to self-report perceived behavioral changes (increased, decreased or no change) compared with early 2021 when the COVID-19 vaccination was not widely available. Other dietary behaviors, for example, using public utensils, cooking food thoroughly and use of dietary supplements, were also asked in the survey.

### 2.3. Ethics

The questionnaire was filled in anonymously. Informed consent was required prior to the survey by clicking the “agree” option to confirm willingness to participate voluntarily in the survey. The online survey was conducted in full agreement with the national and international regulations in compliance with the Declaration of Helsinki (2000). This study was approved by Tsinghua University Health Research Center Ethics Committee (NO. 20210129).

### 2.4. Statistical Analysis

Socio-demographic data were divided into three groups by self-reported vaccination status (fully, partially and not vaccinated) as well as by tertiles of the rate of confidence in the protectiveness of COVID-19 vaccine. Data were presented in percentages for categorical variables and mean with standard deviation in parentheses for continuous variables. Significance tests were performed to compare each characteristic among groups. We used Pearson’s chi-squared test to compare categorical variables. Multiple linear and multiple logistic regressions adjusting for key socio-demographic factors were conducted to evaluate the associations between vaccination status/confidence and dietary practices. For categorical-dependent variables with more than two non-ordinal outcomes (e.g., changes in dietary behaviors: increased, decreased or no change), we selected multinomial logistic regression and chose “no change” as the reference group. In all regression models, not vaccinated and the lowest tertile of vaccine confidence (tertile 1) were used as the reference groups. We calculated body mass index (BMI) for the 4716 participants who filled in weight and height information. Asian cutoffs for weight categorization (underweight < 18.5 kg/m^2^, normal 18.5–23.9 kg/m^2^, overweight 24–27.9 kg/m^2^ and obese ≥ 28 kg/m^2^) were then applied [30]. All statistical analyses were conducted using Stata version 17.0 (StataCorp. 2021. Stata Statistical Software: Release 17. College Station, TX, USA: StataCorp LLC). We considered a *p*-value less than or equal to 0.05 (≤0.05) statistically significant.

## 3. Results

The final analysis included 4873 study participants. We observed high self-reported rate of vaccination coverage in our study population: 82% of participants were fully vaccinated against COVID-19, 14% were partially vaccinated and 4% were not vaccinated (Table 1). The mean score of the perceived protectiveness (vaccination confidence) of the vaccine was 83 (±16) among fully vaccinated participants and lower among unvaccinated participants (mean = 74 ± 22). Among the tertiles of vaccination confidence, people reported a mean score of 98.8 (2.2), 86.3(3.6) and 67.3(14.1) for Tertile 3, Tertile 2 and Tertile 1, respectively (Table 2).

More than half of the included respondents were female (61%), most aged 18 to 45 years (91%), had received bachelor’s or higher degrees (85%), lived in urban areas (84%) and did not report any chronic diseases diagnosed before the COVID-19 pandemic (87%) (Table 1 and Table 2). About 65% of participants were living with vulnerable individuals (e.g., children under the age of 5, elders over the age of 65 or pregnant women) in the same household at the time of survey. Among the 4716 participants who reported weight and height measurements, about 31% were overweight or obese and 13% were underweight (Table 1 and Table 2). The proportions of being female, overweight or obese and well-educated were higher among those who were unvaccinated. Moreover, compared with unvaccinated individuals, those who were fully vaccinated and who had a stronger confidence in the protectiveness of the COVID-19 vaccine reported higher household level dietary diversity (Table 3).

The frequency of dietary intake of the several food groups was found to vary by vaccination status and tertiles of vaccination confidence. The multivariable adjusted models revealed that people who rated the protectiveness of the COVID-19 vaccine the highest (Tertile 3) consumed grains, meat, oil and fats less frequently but fish and seafood, bean and dairy products more frequently than people who believed less in the protectiveness of the vaccine (Tertile 1) (Figure 2a). Some of these differences were also observed between people in Tertile 2 and Tertile 1 (Figure 2b). Moreover, having received the vaccination appeared to additionally associate with reduced vegetable and increased sugar intake frequencies as well as higher frequencies in consuming barbequed, preserved or fried foods (Figure 2c,d).

The odds of extra intake of any supplements or dietary supplements in response to the pandemic were higher among vaccinated (fully or partially) people and people with a stronger confidence in the COVID-19 vaccine (Figure 3a–d). Similar patterns were detected in other cooking and eating behaviors: receiving full or part of the COVID-19 vaccination or having stronger confidence in the protectiveness of the vaccine was positively associated with the odds of adhering to meal splitting, using public utensils, cooking foods thoroughly and separating raw and cooked food (*p* values < 0.001) (Figure 3a–d).

Regarding changes in dietary behaviors, compared to early 2021, a large proportion of the study population indicated a decrease in ordering takeaways (45%) or eating out (52%) and an increase in cooking at home (55%) and online grocery shopping (41%) (Figure 4). While most of these behaviors as well as food consumptions did not differ by vaccination status, full vaccination was associated with an increase (OR = 1.57, *p* = 0.013) in shopping for groceries online and partial vaccination was associated with higher odds of both increasing (OR= 1.86, *p* = 0.003) and decreasing (OR= 1.83, *p* = 0.007) online grocery shopping (Table 4). Changes in food consumption were also evident in the study sample. About 40–50% of people had consumed less seafood, raw food, frozen food and imported fresh produce in August 2021 than earlier the same year. Additionally, about one third of participants increased snack and beverage consumption (Figure 4). It is worth noting that, as shown in the multinomial logistic regression, people stating stronger confidence in the COVID-19 vaccine were more likely to modify their food consumption in 2021, in the direction of either increasing or decreasing (Table 4).

## 4. Discussion

During the COVID-19 pandemic, numerous factors have played roles in influencing and shaping people’s health behaviors. Besides the large-scale lockdowns and restrictions in 2020, COVID-19 vaccine status and confidence could be another major element inducing behavioral modifications, which is supported by results from this study. Most of our participants were fully vaccinated (>80%) by September 2021. The findings from this study revealed distinctive patterns in food consumption, acquisition and eating behaviors among people with different vaccination statuses and varying degrees of confidence in the protectiveness of the vaccine against COVID-19.

### 4.1. Dietary Diversity

Ours results showed that unvaccinated people and people who were not confident in the COVID-19 vaccine reported lower household level dietary diversity than that of vaccinated people and people having a strong belief in the vaccine. Dietary diversity score has been used as an indicator of nutrient adequacy [31,32]. Diverse diets are indispensable in modulating and, potentially, enhancing immune responses [33,34]. On the one hand, receiving the COVID-19 vaccine may have reduced COVID-19-related concerns and stresses that are often associated with unhealthy dietary behaviors [35,36]. A study published in 2021 found that COVID-19-related anxiety was associated with substantially lower dietary diversity [37]. That said, people who were more assured that the vaccine could protect them from infection or severe cases may feel more relieved and less worried after the vaccine has become widely available than people who trusted the vaccine less, therefore reporting improvements in the overall dietary diversity.

On the other hand, people who were not vaccinated by the time of this study could have medical conditions or chronic diseases that prevented them from getting the vaccines. In our study sample, more than 40% of unvaccinated people were overweight or obese and about 10% of them reported the diagnosis of other chronic diseases before the pandemic. These people are likely at higher risk of an impaired immune system and poor diets [38,39]. They may face increased risk of contracting the coronavirus and having severe COVID-19 cases, which could be further aggravated by less diverse diets. Accessing and maintaining healthy and diverse diets is, therefore, of particular importance during the pandemic, especially among the unvaccinated people and people with existing medical conditions.

Intriguingly, among the unvaccinated participants or those who had lower confidence in the COVID-19 vaccine, more were female, with higher education levels and were living in urban areas. Wang et al. conducted a study on vaccination willingness and coverage in China in early 2021 and observed similar demographic associations [40]. This population may possess more knowledge about this disease and may have more access to relevant, though sometimes uncertain and mixed, information on the pandemic. They may exhibit health behaviors including dietary behaviors that are attributable to higher beliefs in themselves and their individual opinions on disease prevention and health management. Despite the emerging literature on vaccination acceptance and hesitancy, very few works have reported the characteristics of the unvaccinated population and their health behaviors. Our study identified several demographic characteristics associated with unvaccinated people and their dietary behaviors, thereby contributing to this field. Further investigations are needed to explore other factors and barriers that potentially negatively impact dietary behaviors including dietary diversity among the unvaccinated population.

### 4.2. Dietary Intake Frequency

A closer look at the dietary intake frequencies of food groups assessed by HDDS unearthed variations in the types of food consumed by people with different vaccination statuses and confidence. Of these, an increase in fish and seafood consumption is evident among vaccinated (fully or partially) people. Previous studies have reported a reduction in fish and seafood consumption, which was possibly associated with safety concerns of virus contamination and supply limits [26,41]. Nonetheless, fish and seafood are known good sources of protein and omega-3 fatty acids and are generally low in calories. They support immune functions and are an essential part of healthy diets [42,43]. Being vaccinated seems to help alleviate safety-related concerns and encourage seafood consumption. In the meantime, it is possible that high vaccine coverage and low COVID-19 cases in August 2021 have allowed seafood supply to recover and become more available to consumers.

Another noticeable trend is the reduction in vegetable, cereal and meat intake among vaccinated people. Moreover, vaccination appears to be associated with a higher frequency of eating unhealthy foods including fried, preserved and barbequed foods. For these who were only partially vaccinated, their consumption of fried, persevered and barbequed food as well as sugar and honey were much higher than unvaccinated people. A plausible explanation is that partially vaccinated people are the least health aware among the three groups (fully, partially and not vaccinated) because those who were fully vaccinated were diligently taking care of their health by getting the full vaccination in time and those who were not vaccinated may take extra caution in their diets to stay healthy. This is in line with the observed socio-demographic characteristics: partially vaccinated people are more likely to be young, male, less educated, earning a lower income and more likely to live in rural areas than the other two groups. Many studies have shown that unhealthy diets (e.g., diets high in fat and refined sugar and low in fiber) are often associated with adverse health outcomes such as weight gain, cardiovascular diseases, cancer and gut dysbiosis as well as impaired immune responses against viruses [44,45,46,47]. Additionally, a prospective cohort study suggested that a diet rich in fruits and vegetables was associated with a lower risk and severity of COVID-19 [48]. Our study results indicate that public health events such as vaccination status could modify people’s behaviors and their impacts are often associated with certain socio-demographic characteristics. Given the current pandemic and high transmissibility of new coronavirus variants, proper diets should be encouraged, regardless of vaccination status, to prevent unhealthy dietary behaviors. Additionally, attention should be paid to people with delayed completion of full vaccination as they are more likely to have poor diets. Diverse strategies may also be considered and employed for different populations.

Level of confidence in the protectiveness of the COVID-19 vaccine also shifted what people were eating, though sometimes to opposite directions than those by vaccination status. Similar to the observations made about vaccination status, people with stronger beliefs in the COVID-19 vaccine reported consuming fish and seafood more frequently and ate less meat and cereal. However, there was no significant increase in unhealthy food consumption (i.e., fried, preserved and barbequed foods). Additionally, interestingly, there was a reduction in oil and fat intake and an increase in fruits, vegetable and dairy product intake. Compared with a single/one-time event such as a vaccination, people’s beliefs may exert more profound and consistent influence on their health behaviors. People with strong confidence in the vaccine could be more health conscious and tend to follow public health guidelines including dietary guidelines and adhere to healthy dietary behaviors.

Modeling heath behaviors often involves numerous personal and environmental factors and can be complicated. Divergent dietary behaviors in response to the pandemic have been reported in the literature [10]. Furthermore, as suggested by the Health Belief Model, a person’s perceived susceptibility to disease/health problems, benefits of the recommended treatment or preventative actions and barriers to obtain the treatment or realize the action are consistent predictors of desired health behaviors [49,50]. Some vaccinated people may feel that COVID-19 is less of a threat and therefore use being vaccinated as an excuse to pay less attention to their diet and allow themselves to indulge in less healthy foods. Meanwhile, grouping people by their level of confidence in the COVID-19 vaccine may have unintentionally selected people who are more health aware into the same group regardless of their vaccination status. Indeed, people having stronger confidence in the effectiveness of the COVID-19 vaccine tend to report a higher consumption of healthy foods such as fish, bean and fruits as mentioned above.

### 4.3. Other Dietary Behaviors and Changes in Comparison to Early 2021

Lastly, the overall impact of the COVID-19 vaccination was assessed by comparing several food shopping and eating behaviors with early 2021 when the number of vaccines administered in China was low [51]. More than one third of the study participants reported an increase in shopping online or at a farmer’s market and cooking at home. High vaccine coverage by September 2021 could have encouraged more in-person grocery shopping. Additionally, individuals and the society have gradually adapted to the new routines during long periods of social distancing and restrictions. Online shopping and delivery services have become more accessible and people working from home have become accustomed to ordering online and cooking at home.

Most changes in dietary behaviors such as eating out, snacking and consuming frozen foods did not vary significantly by vaccination status, except that vaccinated people were more likely to report shopping online for groceries. An online survey collected expected changes in food acquisition activities among 1,000 U.S. consumers in mid-March 2021. Its results indicated that vaccinated people or those planning to be vaccinated could be more risk averse than those who were unvaccinated: people who have received COVID-19 vaccines or plan to get vaccinated were more likely to engage in isolated food acquisition activities such as grocery shopping online and ordering meal services [52]. Moreover, the survey showed that more than half of the respondents did not expect to change their food acquisition activities (e.g., eating at restaurants and shopping for groceries in person) post-vaccination. Both our study and the earlier survey suggested that vaccination status may have a limited impact on how people acquire/purchase foods.

Rather, vaccination belief played a much more important role in impacting how people have changed their dietary behaviors including food consumption and acquisition. People with a stronger belief in the COVID-19 vaccine were more likely to report that they had a dietary behavioral change, either decrease or increase in comparison to early 2021. Likewise, by examining the use of dietary or any supplements during the pandemic, it is obvious that confidence in the COVID-19 vaccination is positively associated with supplement use, and the association is stronger than that by vaccination status.

Individual attitudes and beliefs towards health problems or behaviors could be important predictors of actions. A few studies investigating the intention to receive the COVID-19 vaccine found that perceived benefits of the vaccine and perceived severity of the COVID-19 infection were positively associated with a willingness to get vaccinated [53,54]. Additionally, a randomized field trial study conducted in 2019 showed that several health belief model constructs, such as perceived susceptibility to health problems, were significantly associated with dietary quality [55]. However, the lack of clear directionality of these observed behavioral changes suggests that other elements such as personal health knowledge, comfort level of performing the behavior/activity, perception and skills or other external factors, such as local food availability, should all be considered as possible influential factors on people’s dietary decisions and practices. For example, seafood is typically considered nutritious but requires certain culinary skills to prepare in Chinese cuisine. For people who are health aware and equipped with cooking skills, they may choose to eat more seafoods, whereas for people who are more concerned about the safety of consuming seafood over its nutritional values during the pandemic or people who lack the basic skills to prepare seafood at home may end up eating less of it. In addition, it should be noted that there was a wave of rapidly spreading Delta variants in the Fall of 2021 when this survey was conducted, which may have induced further uncertainty in how people have altered their dietary behaviors.

### 4.4. Limitations

This study was a cross-sectional survey conducted online. We could observe and detect associations but were not able to infer or establish any causality from the study results. Moreover, dietary intake, weight and height measurements were self-reported and thus could have been impacted by recall biases or underestimation. There was only a limited number of food items included in the questionnaire and the actual intake amount was not assessed. Moreover, this study did not evaluate the impacts of food- and nutrition-related knowledge and concerns on food safety, which could have hindered our understanding of behavioral changes during the pandemic. Last but not least, we did not collect reasons for choosing to not get the COVID-19 vaccine. The answers to this question could be potentially valuable in future research in relevant fields and shed light on the determinants of dietary behaviors and the nutritional profile of non-vaccinated people.

## 5. Conclusions

This study explored dietary practices following the COVID-19 vaccine during the COVID-19 pandemic in mainland China. Vaccination status and high confidence were associated with increased household dietary diversity and variations in dietary intakes, with a consistent increase in seafood consumption. However, an increase in the consumption of unhealthy foods such as sugar, preserved and fried food and decrease in fruits and vegetables consumption were also detected among vaccinated people. These dietary intake alterations may have negative health consequences during the ongoing pandemic. We also found that vaccination status and confidence could modulate certain dietary practices differently and they were associated with changes in dietary behaviors compared to early 2021, though no clear directionality could be inferred. Further research is needed to explore influential factors in modeling dietary behaviors during a pandemic. Notably, although some countries have started transitioning out of the pandemic phase recently, the COVID-19 infection still poses a threat to individual and public health. Healthy eating behaviors are imperative and should be encouraged throughout as well as after the pandemic.

## Figures and Tables

**Figure 1 foods-11-01365-f001:**
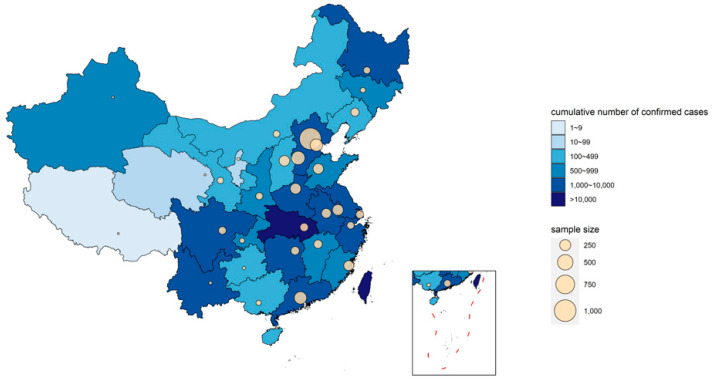
Geographical distribution of participants in the study. The color of the map indicates the cumulative number of confirmed cases in each province by the end of August 2021, according to the report from the Chinese Disease and Control Center [28]. Bubble size is proportional to the sample size of every investigation point.

**Figure 2 foods-11-01365-f002:**
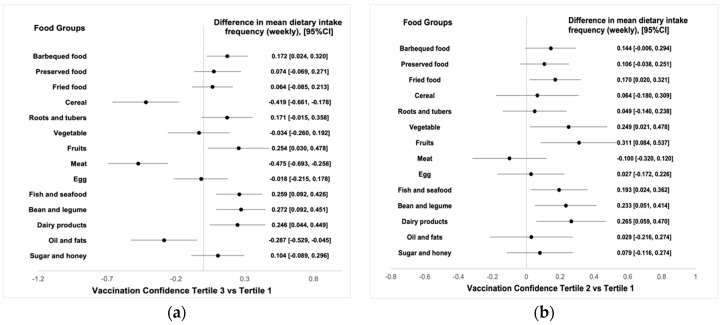
Dietary intake frequency by vaccination status and confidence tertiles. (**a**–**d**): difference in the mean weekly dietary intake frequency in the past month among four comparisons: Tertile 3 vs. Tertile 1 (vaccination confidence), Tertile 2 vs. Tertile 1 (vaccination confidence), fully vs. not vaccinated and partially vs. not vaccinated, respectively.

**Figure 3 foods-11-01365-f003:**
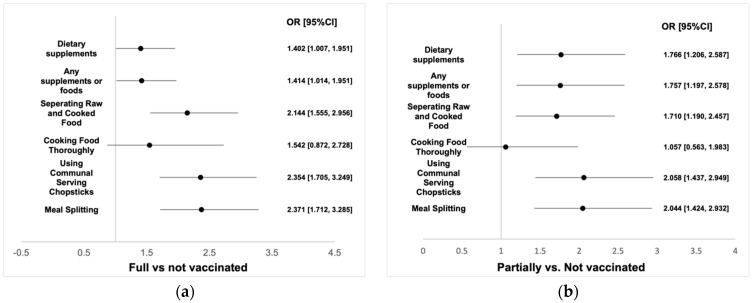
Supplement use and other dietary behaviors by vaccination status and confidence tertiles. (**a**–**d**): odds ratios for four comparisons: fully vs. not vaccinated, partially vs. not vaccinated, Tertile 3 vs. Tertile 1 (vaccination confidence) and Tertile 2 vs. Tertile 1 (vaccination confidence), respectively.

**Figure 4 foods-11-01365-f004:**
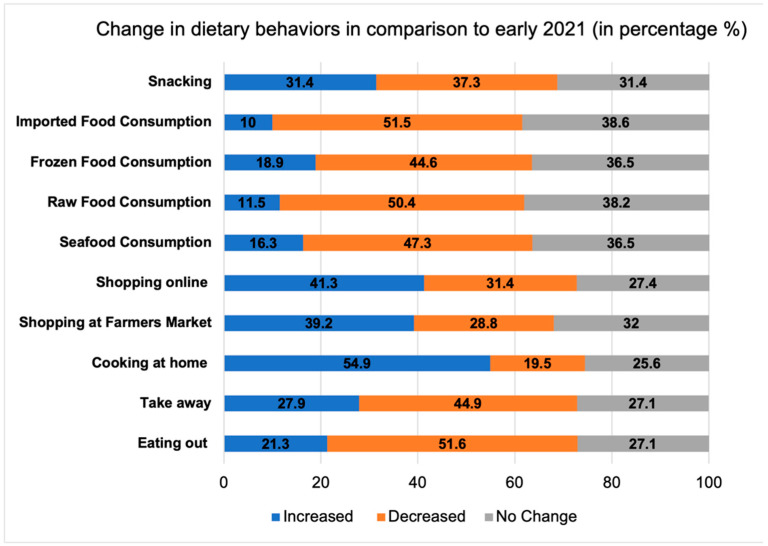
Proportions of people reporting an increase, decrease or no change in dietary behaviors compared to early 2021.

**Table 1 foods-11-01365-t001:** Study participant characteristics by vaccination status.

	Total	Fully Vaccinated	Partially Vaccinated	Not Vaccinated	*p* Value *
	N = 4873	N = 4001	N = 680	N = 192	
Age					<0.001
18–30	55.1%	53.7%	62.9%	56.2%	
31–45	36.3%	37.7%	28.1%	35.9%	
46–60	7.6%	7.7%	7.9%	5.2%	
>60	1.0%	0.9%	1.0%	2.6%	
Sex					<0.001
Male	38.7%	38.3%	46.5%	21.4%	
Female	61.3%	61.7%	53.5%	78.6%	
BMI (N = 4716)					<0.001
Underweight	12.7%	12.4%	15.3%	10.9%	
Normal	53.1%	54.7%	46.3%	44.3%	
Overweight	25.2%	24.8%	24.4%	35.4%	
Obese	5.4%	5.3%	6.3%	5.7%	
Missing	3.5%	2.8%	7.6%	3.6%	
Education level					<0.001
High school or less	15.0%	13.6%	24.1%	11.5%	
Bachelor’s degree	68.7%	70.2%	63.2%	55.7%	
Master’s degree or above	16.3%	16.2%	12.6%	32.8%	
Total annual household income					0.001
<30K	6.2%	5.5%	9.9%	5.7%	
30–100K	27.5%	27.5%	28.1%	26.6%	
100–30K	48.5%	49.6%	42.6%	47.4%	
300–500K	13.1%	13.0%	14.1%	13.0%	
500k–1M	3.6%	3.3%	4.4%	5.7%	
>1M	1.1%	1.1%	0.9%	1.6%	
Diagnosed with chronic disease pre-COVID-19					<0.001
No	86.5%	87.1%	81.6%	90.1%	
Yes	13.5%	12.9%	18.4%	9.9%	
Living with a child ≤ 5 y-o					0.42
No	65.3%	65.6%	65.0%	60.9%	
Yes	34.7%	34.4%	35.0%	39.1%	
Living with elders ≥ 60 y-o					0.011
No	49.9%	50.1%	46.3%	58.3%	
Yes	50.1%	49.9%	53.7%	41.7%	
Living with a pregnant woman					<0.001
No	94.4%	95.6%	91.6%	79.7%	
Yes	5.6%	4.4%	8.4%	20.3%	
Type of residence					<0.001
Rural	16.4%	15.2%	25.3%	8.9%	
Urban	83.6%	84.8%	74.7%	91.1%	

* *p* values were estimated from chi-square test for categorical variables. Values are displayed in percentages (%).

**Table 2 foods-11-01365-t002:** Study participant characteristics by rate of confidence in the protectiveness of the COVID-19 vaccination.

	Total	Tertile 1 67.3 (14.1)	Tertile 2 86.3 (3.56)	Tertile 3 98.8 (2.20)	*p* Value *
	N = 4873	N = 1845	N = 1424	N = 1604	
Age (%)					0.001
18–30	55.1%	58.8%	54.6%	51.1%	
31–45	36.3%	32.8%	36.8%	40.0%	
46–60	7.6%	7.4%	7.6%	7.9%	
>60	1.0%	1.0%	1.0%	1.1%	
Gender (%)					<0.001
Male	38.7%	35.2%	39.8%	41.9%	
Female	61.3%	64.8%	60.2%	58.1%	
BMI (N = 4716)					0.16
Underweight	12.7%	13.6%	12.3%	12.2%	
Normal	53.1%	53.8%	54.8%	50.9%	
Overweight	25.2%	24.3%	24.6%	26.7%	
Obese	5.4%	4.9%	5.3%	6.2%	
Missing	3.5%	3.5%	3.0%	3.9%	
Education level (%)					<0.001
High school or less	15.0%	12.8%	12.5%	19.7%	
Bachelor’s degree	68.7%	63.3%	71.8%	72.1%	
Master’s degree and above	16.3%	23.8%	15.7%	8.2%	
Total annual household income (%)					<0.001
<30K	6.2%	6.3%	4.3%	7.7%	
30–100K	27.5%	26.0%	27.9%	29.0%	
100–30K	48.5%	46.6%	51.8%	47.8%	
300–500K	13.1%	13.8%	12.6%	12.8%	
500k–1M	3.6%	5.7%	2.6%	2.0%	
>1M	1.1%	1.6%	0.8%	0.7%	
Diagnosed with chronic disease pre-COVID-19 (%)					0.018
No	86.5%	85.2%	88.6%	86.0%	
Yes	13.5%	14.8%	11.4%	14.0%	
Living with a child ≤ 5 y-o (%)					<0.001
No	65.3%	69.8%	66.9%	58.7%	
Yes	34.7%	30.2%	33.1%	41.3%	
Living with elders ≥ 60 y-o (%)					<0.001
No	49.9%	54.1%	50.2%	44.8%	
Yes	50.1%	45.9%	49.8%	55.2%	
Living with a pregnant woman (%)					0.004
No	94.4%	93.1%	95.8%	94.6%	
Yes	5.6%	6.9%	4.2%	5.4%	
Type of residence (%)					<0.001
Rural	16.4%	14.1%	15.9%	19.4%	
Urban	83.6%	85.9%	84.1%	80.6%	

* *p* values were estimated from chi-square test for categorical variables. Values are displayed in percentages (%).

**Table 3 foods-11-01365-t003:** Household dietary diversity score by vaccination status and rate of confidence in the protectiveness of the COVID-19 vaccine.

	**Vaccination status**		
	Mean HDDS (SE) *	β (95% CI) #	*p* values
Not vaccinated (n = 192)	9.21 (0.148)	Reference	
Partially vaccinated (n = 680)	9.38 (0.080)	0.170 (−0.161, 0.500)	0.315
Fully vaccinated (n = 4001)	9.54 (0.032)	0.321 (0.024, 0.618)	0.034
	**Confidence in the protectiveness of vaccination against COVID-19**		
	Mean HDDS (SE) *	β (95% CI) #	*p* values
Tertile 1 (n = 1845)	9.22 (0.047)	Reference	
Tertile 2 (n = 1424)	9.57 (0.053)	0.350 (0.210, 0.490)	<0.001
Tertile 3 (n = 1604)	9.77 (0.050)	0.544 (0.407, 0.682)	<0.001

# Multiple linear regression (MLR) model adjusted for age, gender, BMI, education, household income, living with children ≤ 5, living with elders ≥ 65, living with a pregnant woman, having chronic disease before COVID-19, type of residence and quarantine status. * mean HDDS predicted using “margins” command in Stata from the MLR model.

**Table 4 foods-11-01365-t004:** Changes in dietary behaviors in comparison to early 2021.

	Relative Risk Ratio (RRR) (95% CI) # (Reference = No Change)							
	Vaccination Status (Reference= Not Vaccinated)				Belief in the Protectiveness of the Vaccination against COVID-19 (Reference = Tertile 1)			
	Partially Vaccinated (n = 680)		Fully Vaccinated (n = 4001)		Tertile 2 (n = 1424)		Tertile 3 (n = 1604)	
	Increased	Decreased	Increased	Decreased	Increased	Decreased	Increased	Decreased
Eating out	1.251 (0.769, 2.033)	1.447 * (0.982, 2.133)	1.378 (0.896, 2.119)	1.425 ** (1.010, 2.012)	1.184 (0.962, 1.458)	1.335 *** (1.127, 1.582)	1.323 *** (1.074, 1.630)	1.531 *** (1.290, 1.816)
Takeaway	1.218 (0.765, 1.939)	1.060 (0.710, 1.582)	1.370 (0.904, 2.076)	1.144 (0.801, 1.635)	1.309 *** (1.079, 1.588)	1.437 *** (1.205, 1.713)	1.280 ** (1.052, 1.557)	1.607 *** (1.349, 1.915)
Cooking at home	1.150 (0.777, 1.704)	1.884 ** (1.112, 3.191)	1.157 (0.819, 1.634)	1.481 (0.916, 2.393)	1.530 *** (1.289, 1.814)	1.185 (0.955, 1.470)	1.902 *** (1.597, 2.266)	1.374 *** (1.104, 1.709)
Shopping at a farmer’s market	1.445 * (0.963, 2.169)	1.639 ** (1.066, 2.520)	1.249 (0.872, 1.790)	1.237 (0.842, 1.816)	1.549 *** (1.304, 1.839)	1.179 * (0.981, 1.416)	1.975 *** (1.661, 2.348)	1.413 *** (1.174, 1.700)
Shopping online	1.858 *** (1.239, 2.786)	1.825 *** (1.175, 2.835)	1.565 ** (1.097, 2.232)	1.408 * (0.951, 2.087)	1.417 *** (1.190, 1.687)	1.258 ** (1.040, 1.521)	1.494 *** (1.252, 1.782)	1.451 *** (1.204, 1.750)
Seafood consumption	1.321 (0.790, 2.211)	1.367 (0.938, 1.991)	1.390 (0.876, 2.204)	1.299 (0.927, 1.821)	1.261 ** (1.019, 1.561)	1.405 *** (1.196, 1.650)	1.366 *** (1.102, 1.693)	1.669 *** (1.422, 1.960)
Raw food consumption	1.777 * (0.919, 3.437)	1.225 (0.853, 1.761)	1.716 * (0.932, 3.162)	1.160 (0.840, 1.602)	1.183 (0.924, 1.514)	1.409 *** (1.205, 1.648)	1.246 * (0.978, 1.588)	1.567 *** (1.341, 1.832)
Frozen food consumption	1.521 * (0.945, 2.447)	1.360 (0.923, 2.003)	1.227 (0.798, 1.886)	1.324 (0.937, 1.872)	1.324 *** (1.083, 1.618)	1.475 *** (1.251, 1.738)	1.411 *** (1.150, 1.732)	1.892 *** (1.606, 2.229)
Imported produce consumption	1.923 * (0.928, 3.986)	1.379 * (0.962, 1.975)	1.855 * (0.938, 3.668)	1.362 * (0.988, 1.877)	1.340 ** (1.029, 1.744)	1.479 *** (1.266, 1.727)	1.432 *** (1.107, 1.853)	1.708 *** (1.462, 1.995)
Snack and beverage consumption	1.366 (0.876, 2.130)	0.816 (0.545, 1.222)	1.275 (0.853, 1.905)	0.866 (0.605, 1.240)	1.278 *** (1.066, 1.532)	1.188 * (0.997, 1.416)	1.353 *** (1.127, 1.623)	1.331 *** (1.118, 1.583)

*** *p* < 0.01, ** *p* < 0.05, * *p* < 0.1. # Multinomial logistic regression model adjusted for age, gender, BMI, education, household income, living with children ≤ 5, living with elders ≥ 65, living with a pregnant woman, having chronic disease before COVID-19, type of residence and quarantine status.

## Data Availability

Data used in this report may be available upon requests made to the corresponding author.

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
