# Peer review of "Impact of COVID-19 Vaccination Status and Confidence on Dietary Practices among Chinese Residents"

_foods, 2022, doi:10.3390/foods11091365_

Round 1
Reviewer 1 Report
The study aimed to investigate the impacts of vaccination status and confidence on dietary practices. The methodology is a quantitative one, an online survey being conducted in August 2021. The data ere collected on dietary intake, diversity and behaviors, vaccination status and confidence, and socio-demographic characteristics. The results suggest that vaccination status and confidence might be significant influencing factors affecting people’s health behaviors and highlight that healthy eating should be consistently promoted to prevent poor dietary practices during global health crisis.
The study is well conducted. The literature review is complete and new, the statistical analysis is robust
The Conclusion section should be expanded and the authors should stress the new contribution they added through their study at the existing literature.
Author Response
Point 1: The Conclusion section should be expanded and the authors should stress the new contribution they added through their study at the existing literature.
Thank you for the comment and suggestion. We have added a paragraph with an emphasis on our study’s contribution to the literature and a few more sentences in the discussion and the conclusion sections to expand the interpretation of our study findings.
Reviewer 2 Report
Dear Authors,
the research topic is interesting. The Covid-19 pandemic is new to the world, and it is worth researching people's behavior at this time. I rate the paper as good, but in order to improve it, I suggest some additions.
Editorial corrections:
Lines 311 and 472: suggest During in place of Amid;
Line 325: here is [30] - [31], should be [30, 31];
Substantive suggestions:
1) Introduction - in my opinion it would be nice to mention the possible reasons for the distrust of Covid vaccines as this is included in the title. Maybe there are data on what percentage of people do not trust, what are their characteristics.
2) Subtitle 2.4. Line 156-157- it would be nice to provide a scientific basis for BMI categorization.
3) Subtitle 4.4. In the limitations it would be good to mention that the reasons for not being vaccinated were not asked. The answers to this question could be used to indicate the need for further research. Perhaps this could shed light on the determinants of the nutritional profile of non-vaccinated people.
Author Response
Thank you for your comments and suggestions. We found them very helpful. Please see our responses below:
Point 1 - Lines 311 and 472: suggest During in place of Amid
Response: We have replaced “amid” with “During”
Point 2 - Line 325: here is [30] - [31], should be [30, 31];
Response: We have re-created the in-text citation and they are enclosed by same brackets.
Point 3 - Introduction - in my opinion it would be nice to mention the possible reasons for the distrust of Covid vaccines as this is included in the title. Maybe there are data on what percentage of people do not trust, what are their characteristics.
Response: we agree that it would be interesting to explore reasons for vaccination hesitancy and the percentage of people who do not trust the vaccine and their characteristics. We did find a few relevant papers and cited one (reference 40) in our manuscript. However, the focus of this article is more on the impacts of vaccination status and confidence on dietary behaviors but less on people’s perceptions of vaccination. We think that the introduction section may not be the best fit for the suggested content and therefore added a few relevant sentences in the discussion section.
Point 4 - Subtitle 2.4. Line 156-157- it would be nice to provide a scientific basis for BMI categorization.
Response: we added the reference where these cut-offs were based on.
Point 5 - Subtitle 4.4. In the limitations it would be good to mention that the reasons for not being vaccinated were not asked. The answers to this question could be used to indicate the need for further research. Perhaps this could shed light on the determinants of the nutritional profile of non-vaccinated people.
Response: yes, this is a very good point. We added it to the limitation section.